# Spatio-Temporal Distribution, Spillover Effects and Influences of China’s Two Levels of Public Healthcare Resources

**DOI:** 10.3390/ijerph16040582

**Published:** 2019-02-17

**Authors:** Xueqian Song, Yongping Wei, Wei Deng, Shaoyao Zhang, Peng Zhou, Ying Liu, Jiangjun Wan

**Affiliations:** 1School of Management, Chengdu University of Information Technology, Chengdu 610225, China; sxq@cuit.edu.cn; 2School of Earth and Environmental Sciences, The University of Queensland, Brisbane, QLD 4067, Australia; 3Research Center for Mountain Development, Institute of Mountain Hazards and Environment, Chinese Academy of Sciences, Chengdu 610041, China; zhangsyxs@163.com (S.Z.); zhoupeng726@sina.com (P.Z.); liuying@imde.ac.cn (Y.L.); 4College of Resources and Environment, University of Chinese Academy of Sciences, Beijing 100049, China; 5Department of Urban and Rural Planning, School of Architecture and Urban-Rural Planning, Sichuan Agricultural University, Chengdu 610041, China; wanjiangjun@outlook.com

**Keywords:** two levels of healthcare resources, spatial spillover effects, spatial equity, dynamic spatial Durbin panel model

## Abstract

In China, upper-level healthcare (ULHC) and lower-level healthcare (LLHC) provide different public medical and health services. Only when these two levels of healthcare resources are distributed equally and synergistically can the public’s demands for healthcare be met fairly. Despite a number of previous studies having analysed the spatial distribution of healthcare and its determinants, few have evaluated the differences in spatial equity between ULHC and LLHC and investigated their institutional, geographical and socioeconomic influences and spillover effects. This study aims to bridge this gap by analysing panel data on the two levels of healthcare resources in 31 Chinese provinces covering the period 2003–2015 using Moran’s *I* models and dynamic spatial Durbin panel models (DSDMs). The results indicate that, over the study period, although both levels of healthcare resources improved considerably in all regions, spatial disparities were large. The spatio-temporal characteristics of ULHC and LLHC differed, although both levels were relatively low to the north-west of the Hu Huanyong Line. DSDM analysis revealed direct and indirect effects at both short-and long-term scales for both levels of healthcare resources. Meanwhile, the influencing factors had different impacts on the different levels of healthcare resources. In general, long-term effects were greater for ULHC and short-term effects were greater for LLHC. The spillover effects of ULHC were more significant than those of LLHC. More specifically, industrial structure, traffic accessibility, government expenditure and family healthcare expenditure were the main determinants of ULHC, while industrial structure, urbanisation, topography, traffic accessibility, government expenditure and family healthcare expenditure were the main determinants of LLHC. These findings have important implications for policymakers seeking to optimize the availability of the two levels of healthcare resources.

## 1. Introduction

Healthcare policymakers optimise the location and organisation of public healthcare resources according to a trade-off between spatial equity and cost-effectiveness. China has achieved remarkable success in the medical and health service sectors during the last three decades [1]. However, in the current healthcare system in China, patient access to healthcare services is not organised according to a gatekeeping system and two-directional referral network. Therefore, due to the unordered pattern of medical treatment, upper-level hospitals are always overcrowded, while lower-level health centres have fewer patients. This situation increases medical costs, wastes healthcare resources and lowers healthcare efficiency. To solve this problem, to develop hierarchical diagnosis and treatment (HDT) system initiated in 2015 has become a main objective of the Chinese healthcare reform [2]. It emphasises that residents’ different medical service demands should correspond to the different levels of medical institutions, and the main functions of different levels of healthcare resources should be unambiguously divided. The upper-level healthcare (ULHC) should concentrate on illness and disease treatment, public health services, and scientific research. The lower-level healthcare (LLHC) perform first diagnoses, rehabilitation therapy and basic public health services [3,4]. Only when these two levels of healthcare resources are distributed equally and synergistically can investment in public health be utilised efficiently and people’s demands for healthcare met fairly. With this background, it is important to understand the spatial equity of the two levels of healthcare resources across China.

The spatial equity (or inequity) of healthcare resources, ranging from healthcare professionals to healthcare institutions, has been analysed. The analysing methods have been developed from simple economic index to complex spatial data models. For example, the spatial disparity between physicians in hospitals and clinics and the population in Japan was estimated by Lorenz curves and Gini coefficients [5]. Disparity in the spatial distribution of clinics within the city of Daejeon was analysed by hot-spot analysis [6]. Geographic distribution of healthcare resources in China was estimated by dynamic convergence model [7]. Spatial disparities of access to primary healthcare across rural Australia have been revealed by a modified two-step floating catchment area method [8]. While previous studies have contributed to understanding healthcare spatial disparity, few have evaluated the differences in spatial equity in different levels of healthcare resources.

It is argued that the distribution of healthcare resources is influenced by a combination of natural and socioeconomic factors [9,10,11]. Socioeconomic factors such as the economy [7], healthcare investment [12], education [13], government policies [1,14], urbanisation [15] and demographic characteristics [16] are considered important determinants of healthcare resource distribution. For example, Lee pointed out that the population proportion aged over 65 years, the number of businesses and employees contribute to heterogeneity in the spatial distribution of clinics [6]. Qin and Hsieh found that GDP per capita has a significant and non-linear impact on the convergence rate of healthcare resources [7]. Coleman found that access to healthcare in the United States is limited by financial, organisational, social and cultural barriers [17]. Bhattacharjee et al. argued that the spatial structure of socioeconomic characteristics and health behaviours, and the utilisation and quality of healthcare, are particularly relevant in the efficient allocation of healthcare resources [18]. While recent studies have considered the contribution of natural factors, research has concentrated on environmental variables including wastewater and air pollution emissions [13,19]. However, few quantitative studies have explored how topographical factors influence the spatial distribution of healthcare resources.

A number of researchers have suggested that policymakers should take spatial independence into consideration when aiming to mitigate public service inequality [13,20,21]. Existing studies have focused on the spatial spillover effects of public infrastructure on regional productivity, of fiscal investment on public provision [22,23], and of health investment on regional healthcare costs [24]. Spatial spillover analysis techniques have included the use of cross-sectional and spatial panel data [25,26,27], and static and dynamic models [28,29,30,31]. New approaches have been implemented following empirical studies of public services. For example, Zafra-Gómez and Chica-Olmo analysed spatial panel data on waste collection services in small and medium-sized municipalities in Spain using the spatial autoregressive regression model (SAR) and the spatial Durbin model (SDM) [32]. Mourao and Vilela analysed the multiplier effects of pensions in Portuguese municipalities using the dynamic spatial Durbin model (DSDM) [33]. Empirical evidence on healthcare spatial interdependence strongly suggests that there are spatial spillover effects across regions [13,22,24,34,35]. Quadrado et al. analysed the spatial spillover of health facilities by Theil’s second measure [36]. Mobley et al. used SAR to explain the neighbourhood peer effect in preventive care utilization [37]. Costa-Font and Moscone estimated interdependence in the health spending decisions of neighbouring regions by the spatial lag model (SLM) and spatial error model (SEM) methods [38]. Turi and Grigsby-Toussaint used SDM to estimate the direct and indirect effects of socio-ecological determinants on diabetes-related mortality [39]. Tabb et al. assessed the spillover effects of health factors on health outcomes across the United States by applying SDM [40]. These studies highlight how useful these methods are in exploring healthcare determinants and their spatial spillover effects. Given its externality, the intervention of hierarchical healthcare allocation is much more complicated. However, the spillover effects of natural and socioeconomic characteristics on healthcare resources have not received much scholarly attention. In particular, research on the impacts of such determinants on different levels of healthcare resources is rare and, to our knowledge, dynamic spatial analysis methods have not been applied in healthcare resource distribution research. Thus, these studies may have limited implications for the development of spatial equity and the optimisation of different levels of healthcare resources.

The purposes of this paper are to: (1) analyse the spatio-temporal distribution of ULHC and LLHC and; (2) understand the institutional, geographical and socioeconomic factors influencing ULHC and LLHC distributions and their short-term, long-term, direct and spillover effects in China. It is expected that the findings from this study will help design policy interventions and allocate resources to enhance spatial equity.

## 2. Methods

### 2.1. Defining the Variables of Interest

#### 2.1.1. Defining the Two-Level Healthcare Index

The *Planning Outline of the National Medical and Health Service System (2015–2020)* was issued by the General Office of the State Council of China. According to this document, the public healthcare system is composed of public hospitals (above the county level), professional public health institutions (above the county level) and primary healthcare centres (below the county level). The main functions are defined as medical treatment (for public hospitals) and public healthcare services (for professional public health institutions). These two functions are combined for primary healthcare centres. According to the Planning Outline and publicly-available data and for the purposes of this study, public hospitals include general hospitals, traditional Chinese medicine (TCM) hospitals and specialized hospitals. Public professional healthcare service institutions include control disease centres (CDCs), specialised disease prevention centres (SDPs) and maternal and child health stations (MCHSs). Primary healthcare centres consist of urban community health institutions (UCHIs) and town health centres (THCs) (community level) and village clinics (village level). Most village clinics are operated by private rural doctors and the focus of this study is on public healthcare resources. Therefore, in this study, ULHC refers to the healthcare resources of public hospitals and professional public health institutions, while LLHC refers to the healthcare resources of UCHIs and THCs (Figure 1).

Health professionals, beds and institutions are the most significant criteria for evaluating healthcare resources [13]. Based on the healthcare systems in China as well as relevant research [41], evaluation index system of the two levels of public healthcare resources was established. (Table 1).

#### 2.1.2. Choosing the Factors Influencing Healthcare Resource Distribution

Selection of the factors that may influence healthcare resource distributions was based on relevant literature reviews and data availability. In terms of socioeconomic factors, the non-agricultural industry rate was selected as a proxy indicator of the state of the regional economy, which could have positive impacts on the healthcare resources of a local region [7]. The urbanisation rate was an indicator which could have positive impacts on local and neighbouring regions [42]. Geographical factors primarily included traffic accessibility and the proportion of mountainous area. Traffic accessibility reflects economic mobility and connectivity between regions, which could have positive impacts on healthcare resources in a province and its surrounding provinces [43]. The proportion of mountainous area reflects topography, which was assumed to have negative impacts on local healthcare resources, because there is undeveloped social and economic potential in the mountainous regions of China [4,44]. Government investment includes government healthcare investment and education investment. The former could have positive impacts on healthcare resources in local and surrounding regions due to the competitive and mimetic effects of government fiscal spillover [22,23,45], while the latter could have negative impacts on healthcare resources due to crowding-out effects between different types of public investment [13]. Family healthcare expenditure included urban and rural family healthcare expenditure per capita, which could have positive impacts on local and surrounding healthcare resources. The above 15 independent variables are shown in Table 2.

#### 2.1.3. Data Sources

This study considered 31 provinces, independent municipalities and autonomous regions as spatial units and 2003–2015 as temporal units of analysis. Three types of data were used in this article. (1) Healthcare data. This data included the number of healthcare institutions, hospital beds and healthcare professionals of public hospitals, CDCs, MCHSs, SDPs, UCHIs and THCs. All the data were extracted from the China Health Statistics Yearbooks (CHSY) 2004–2016 provided by the National Health Commission. It should be noted that databases that provide healthcare resource estimates for Chinese provinces are not abundant. For example, the numbers of public hospitals of traditional Chinese and western medicine and public ethnic hospitals were not publicly available. An index system (Table 1) was established according to the availability of public data. (2) Socioeconomic data. These indicators include the non-agricultural industry rate, urbanisation rate, healthcare and education investment, urban and rural family healthcare expenditure per capita and traffic data, such as the lengths of first- and second-class roads, other roads and railways. The resident population at year-end and the numbers of administrative areas were also collected to calculate healthcare resource indices. All the socioeconomic data were extracted from the China Statistical Yearbooks (CSY) (2004–2016) provided by the National Bureau of Statistics of China (http://data.stats.gov.cn/). (3) Geographical spatial data. The proportions of mountainous area in each province were obtained from the Digital Mountain Map of China (DMMC) [46]. This digital map was designed by the Institute of Mountain Hazards and Environment, Chinese Academy of Sciences. It was compiled using digital geomorphology methods and remote sensing data and is used as a reference map for research in mountainous areas. Table 3 lists the descriptive statistics and sources of these variables.

### 2.2. Methods

There were four steps of analysis used in this study. Firstly, the values of ULHC and LLHC were calculated using the entropy method; secondly, the spatio-temporal distribution of ULHC and LLHC were estimated by Moran’s *I* model; thirdly, the stages of the study period were divided by applying mathematical derivative models; finally, the influencing factors and spillover effects were analysed by dynamic spatial panel models.

#### 2.2.1. Calculation of ULHC and LLHC and the entropy method

The numbers of institutions, beds and healthcare professionals for each type of healthcare institution (A) are presented in Figure 1. These were summarized respectively to obtain the total number of each index at the criterion layer. The PONMHSS (2015–2020) proposes that healthcare institutions should be allocated according to administrative areas and service radii, while hospital beds and health professionals should be allocated according to the population. Thus, values of the criterion layer (B) were calculated according to the number of healthcare institutions per 100 km^2^, and the numbers of hospital beds and health professionals calculated per 1000 people. Then, values of *Y* were calculated by the entropy weighted summarization of dimensionless values of B. The entropy method has been widely used in multiple objective comprehensive evaluation research, as it is objective in weighting indexes according to information contained in the data [47]. In this study, there were *n* provinces and *m* indicators. The entropy of the *l*th indicator is expressed as $H_{l}=-k\overset{n}{\underset{i=1}{\sum }}\left(f_{il}lnf_{il}\right),\;i=1,2,\ldots ,n;\;l=1,2,\ldots ,m$, where *f_il_* is the frequency of the *i*th evaluating object in the *l*th indicator, $f_{il}=r_{il}/\overset{n}{\underset{i=1}{\sum }}r_{il}$, $r_{il}\;\mathrm{is}\;\mathrm{the}\;\mathrm{normalized}\;\mathrm{values}\;\mathrm{of}\;B,$ and *k* = 1/ln*n*; if *f_il_* = 0, *f_il_* ln *f_il_* = 0. Then, *w* of the *l*th indicator can be expressed as $w_{l}=\frac{1-H_{l}}{m-\sum_{l=1}^{m}H_{l}}$, where 0 ≤ *w_l_*≤ 1 and $\overset{m}{\underset{l=1}{\sum }}w_{l}=1$. The value of *Y* (ULHC and LLHC) at the goal layer was then calculated by the formula $Y_{i}=\overset{m}{\underset{l=0}{\sum }}w_{l}B_{il}$.

#### 2.2.2. Moran’s I model

Spatial autocorrelation correlates variables with spatial locations and reflects the degree of spatial dependence between values of random variables in geographic terms. The global Moran’s *I* method has been widely used to reflect the degree of spatial autocorrelation of variables and estimate the spatial agglomeration and divergence distribution. It was used to measure the spatio-temporal distributions of the ULHC and LLHC based on Equation (1):(1)$$I=\frac{n\sum_{i=1}^{n}\sum_{j=1}^{n}W_{ij}\left(Y_{i}-\bar{Y}\right)\left(Y_{j}-\bar{Y}\right)}{\sum_{i=1}^{n}\sum_{j=1}^{n}W_{ij}{\left(Y_{i}-\bar{Y}\right)}^{2}}$$where, *Y_i_*, *Y_j_* are the values of the healthcare resource indices of the *i*th and *j*th provincial units, $\bar{Y}$ is the mean of the variable, and *W_ij_* is a matrix of spatial weights. The global Moran’s *I* value range is [–1,1]. The positive value indicates the spatial agglomeration distribution, and the negative value indicates the spatial divergence distribution. Higher Moran’s *I* values indicate stronger spatial structure, and lower Moran’s *I* values indicate weaker spatial structure. Zeroes represent random spatial distributions, i.e., no spatial correlation.

#### 2.2.3. Mathematical Derivative Models

Based on the results of the spatial autocorrelation analysis of ULHC and LLHC, mathematical derivative models were applied to divide the stages of the study period. The first-order backward difference quotient was used for the previous year using the formula $:\;f^{\prime }(I_{t})=\frac{f^{\prime }(I_{t})-f^{\prime }(I_{t-1})}{I_{t}-I_{t-1}}$, where *I* is the Moran index, *t* is the time period (years 2003–2015, thus: *t* = 1, 2, …, 13). The first-order forward difference quotient was applied for the other 12 years using the formula: $f^{\prime }(I_{t})=\frac{f^{\prime }(I_{t+1})-f^{\prime }(I_{t})}{I_{t+1}-I_{t}}.$

#### 2.2.4. Dynamic Spatial Panel Models

To eliminate the heteroscedasticity of the regression model, the logarithms of the variables were used. Stepwise linear regression was applied to avoid the effect of multicollinearity between the explanatory variables. Following the strategy described in LeSage and Pace [48] and Elhorst [49], this study started with the spatial Durbin model (SDM) as a general specification and test for the exclusion of the spatial autoregression model (SAR) and spatial error model (SEM) using likelihood ratio (LR) tests. In this study, we wanted to estimate the long-term and short-term, and direct and indirect effects of changes in geographical, socioeconomic and healthcare expenditure characteristics on ULHC and LLHC. This led to Equation (2), which is empirically associated with a DSDM [28]:(2)$$Y_{it}=\tau Y_{i,t-1}+\delta WY_{i,t}+\eta WY_{i,t-1}+\beta X_{i,t}+\theta WX_{i,t}+\mu +\varepsilon_{i,t}$$where *Y_i,t_* denotes an *N* × 1 vector consisting of one observation of the healthcare resource for every province *i* in the sample at time *t*; *X_i,t_*, and *WX_i,t_* are matrices of exogenous dimensions; *β* and *θ* are the response parameters to these exogenous dimensions;, *τ* and *δ* is the response parameter of the lagged local healthcare resources in time and in space, *η* is the space-time parameter; *μ* represents spatial fixed effects; and *ε_i,t_* represents an error term uncorrelated with the explanatory variables across provinces and over time.

To make the models more robust, Lagrange multiplier (LM) tests were used to select the most appropriate spatial econometric model out of SAR, SEM and SDM. Then, Hausman’s specification test was performed to decide whether fixed effects (FE) or random effects (RE) models would be more appropriate. Following Elhorst, the bias-corrected quasi-maximum likelihood (QML) approach described by Yu, de Jong and Lee was applied to select the appropriate model out of time-lagged, spatio-temporal-lagged, and both time and spatio-temporal-lagged options. The final specification of the dynamic spatial panel data model was used to probe the relationship between a province’s healthcare resources and its explanatory variables, both within the province as well as in neighbouring provinces. All computations are performed by STATA software (Stata Corporation, College Station, TX, USA, 2015).

## 3. Results

### 3.1. Spatio-Temporal Distributions of the Two Levels of Healthcare Resources

#### 3.1.1. Distribution of the Two Levels of Healthcare Resources

The average of 13 years of data was ranked to understand the spatial distribution of ULHC and LLHC (Figure 2). The results indicate, in terms of overall healthcare resources, that Shanghai, Beijing, Tianjin, Zhejiang and Jiangsu were the five highest-ranked provinces, while Xizang, Yunnan, Qinghai, Gansu and Ningxia were the lowest five. In terms of LLHC, the five highest-ranked provinces were Shanghai, Zhejiang, Beijing, Jiangsu and Tianjin, while Ningxia, Heilongjiang, Qinghai, Yunnan and Xizang were the lowest. In terms of ULHC, the five highest-ranked provinces were Shanghai, Beijing, Tianjin, Liaoning and Shandong, while Xizang, Guangxi, Gansu, Yunnan and Guizhou were the lowest five. In summary, these results show that distribution of the LLHC tended to be higher in the eastern, central and southern regions, while that of ULHC were higher in the eastern regions. Both levels of healthcare resources were relatively low north-west of the Hu Huanyong Line, except for Xinjiang province.

#### 3.1.2. Spatio-Temporal Disparity of the Two Levels of Healthcare Resources

High spatial autocorrelations were found to exist for both levels of healthcare. The spatial disparity of ULHC (Moran’s *I*: Max = 0.5559, Min = 0.2884, *p* ≤ 0.01) was more significant than that of LLHC (Moran’s *I*: Max = 0.3212, Min = 0.2134, *p* ≤ 0.01) and the spatial distribution of both levels together tended to be more equal. Based on the derivative value of the Moran’s *I* of both levels of healthcare, the study period was divided into three stages (Figure 3). In the first stage (2003–2005), the spatial disparities increased simultaneously. The two levels of healthcare resources clustered rapidly and both reached the peaks of aggregation in 2005. In the second stage (2006–2009), the spatial disparity of ULHC decreased quickly and the gap in spatial equity between ULHC and LLHC narrowed. In the third stage (2010–2015), both levels of healthcare resources gradually became distributed more equally. The disparity of ULHC decreased slower than that of LLHC. Thus, the gap in spatial equity between the two levels stabilized.

Figure 4 compares the spatial distribution between LLHC and ULHC in each period. Although the two levels of healthcare had been improved greatly during these 13 years, the spatial disparity was still significant. Since 2009, the growth rate of LLHC in western regions had been much higher than those in the eastern coastal regions. But the opposite was true for ULHC; the growth rate in the developed eastern regions had been extremely high. These results indicate that LLHC tended to be distributed more equally than ULHC.

### 3.2. Spillover Effects of the Two Levels of Healthcare Resources and their Determinants

According to the results of stepwise linear regression, the variance inflation factors (VIF) were all less than 10, which suggests that there was no effect of multicollinearity between the explanatory variables. Table 4 presents the constraint statistic for the selection of the spatial econometric models.

The results of LM tests indicate that both LM-Lag and LM-Error, and their robust counterparts, were significant for both ULHC and LLHC. Therefore, the DSDM was selected as the most suitable spatial econometric model. Furthermore, in terms of ULHC, LR-spatial-lag was 40.773 and LR-spatial-error was 51.295. In terms of LLHC, LR-spatial-lag was 43.012 and LR-spatial-error was 65.366. These four constraint statistics all passed 1% significance tests, which indicates that the DSDM cannot be simplified as a spatial lag model or spatial error model. Moreover, this panel data contained data from all provinces in China, rather than being a random sample and, thus, the FE models were generally more appropriate than the RE models [28]. In addition, the results of the Hausman test show that the coefficients of ULHC and LLHC were 108.99 and 22.33, respectively. Both were significant at the 0.01 level, which further confirms that FE models should be used for the final models. The results of QML tests indicate that of the three types of DSDM models, the spatio-temporal lagged models are the best for both levels of healthcare. Therefore, we selected the spatio-temporal-lagged DSDM with FE to test the factors that influence healthcare resources and analyse their spillover effects. Table 5 illustrates the estimates of spatio-temporal-lagged DSDM regression results.

Table 6 reports the results of the DSDMs of ULHC and LLHC. The results of the ULHC model show that the non-agricultural industry rate had a positive direct impact and a negative indirect impact, both in the short- and long-term. Traffic accessibility had a positive direct impact in the short- and long-term, and a positive indirect impact in the long-term but an insignificant indirect impact in the short-term. Government healthcare investment had a substantial positive direct and indirect impacts in both short- and long-terms. On the contrary, the impacts of government education investment were all significantly negative. Urban family healthcare expenditure had a negative direct impact, while rural family healthcare expenditure had positive indirect impacts.

For the LLHC model, the spillover effects were smaller than those for ULHC. The non-agricultural industry rate had a positive direct impact in both the long- and short-term. The direct long- and short-term impacts of urbanization rate turned out to be positive. The proportion of mountainous areas had a negative direct impact in the short-term. Traffic accessibility had significant positive direct impacts with insignificant spillover effects. Government healthcare investment had a positive spillover effect, while the influence was lower than that for ULHC. Similar to ULHC, the negative impacts of government education investment on LLHC were all significant. Urban family healthcare expenditure had positive direct and negative indirect impacts, but the impacts of rural family healthcare expenditure were all insignificant.

Taken together, these results suggest that geographical and socioeconomic factors and healthcare expenditure affect the two levels of healthcare resources differently. Long-term effects are greater for ULHC and short-term effects are greater for LLHC. Moreover, in general, the spillover effects of ULHC are more significant than those of LLHC. This may be explained in terms of different service radii and residents’ spatial behaviours in relation to the different levels of healthcare [50]. LLHC mainly serves community residents, while ULHC usually also services residents living in surrounding jurisdictions. Many patients travel long distances to obtain the best medical treatment for rare or serious illnesses. It should be noted that consumers usually go with their registered providers for planned immunization and maternal and child health care if they want to enjoy governmental subsidies but spillover effects also exist given the large interprovincial mobility of the population in China.

## 4. Determinants of the Two Levels of Healthcare Resources

### 4.1. Healthcare Reform Policies

Three healthcare reforms are the main influences on the three stages of spatial disparity in ULHC and LLHC in China (Figure 3). In 1996, the Chinese central government determined the guidelines and basic principles of health work, emphasising that disparities in healthcare among regions should be reduced. However, the subsequent market-oriented redistribution led to an over-clustering of healthcare resources. Spatial disparities were still large, and they peaked in 2005 due to the lagged effect of policy implementation. Thus, the first stage of spatial disparity in ULHC and LLHC showed a typical increasing trend. In the second stage, spatial aggregation of both levels of healthcare fluctuated, while the spatial disparity of ULHC was reduced obviously since 2007. The main reason for this may be because institutional healthcare reform was initiated in 2006 by the Chinese State Council. In the third stage, the healthcare reforms officially launched in 2009 focused on the provision of affordable and equitable basic healthcare for the masses [2,51,52]. From then on, the spatial equity of LLHC was enhanced markedly. However, although the spatial disparity of UPHC decreased from a global perspective, aggregation in local areas was more concentrated (Figure 4).

China is a country with a large population, varied topography, and rapidly changing social and economic structures. The structural differences between regions remain stark and the mobility of resources strengthens the dependence of regions. Thus, it is essential to further our analysis to understand whether geographical and socioeconomic factors affect healthcare resource distribution in local and neighbouring regions [3].

### 4.2. Geographical Factors

China is a country of vast territory and complicated topography; 94% of its residents live south-east of the Hu Huanyong Line, in 43% of the country’s territory [4]. A worthy subject of investigation is how to promote healthcare equity so that residents of the central and western mountainous areas can also benefit from a modern healthcare system. Previous studies have found that the provision of public goods, including healthcare, is comparatively lower in mountainous areas of China [4,44]. This study confirmed that the proportion of mountainous areas has a negative impact on LLHC. Combining this result with the spatio-temporal distribution analysis (Figure 2, Figure 3 and Figure 4), although LLHC was distributed much more equally than ULHC, there are still structural imbalances. LLHC are highly concentrated in eastern coastal regions with less mountainous areas. On contrary, the DSDM results indicate that ULHC distribution wasn’t significantly influenced by topography.

Traffic accessibility is considered a very important factor that promotes the development of regional economies. Thus, it is no surprise that traffic accessibility had significant positive impacts on both levels of healthcare within a specific province. Since it is well known that the improvement of transport systems promotes mobility and connectivity within and between regions, traffic accessibility was expected to strengthen the interdependence of healthcare resources between regions. It is somewhat surprising that, in this case, the spillover effect of traffic accessibility was only significant on ULHC in the long term. A possible explanation may be that the provinces of China cover very large areas (average = 305,000 km^2^); thus, traffic improvement in one province may only have limited and time-lagged influences on the healthcare resources of neighbouring provinces.

### 4.3. Socioeconomic Factors

Previous studies have emphasised the impact of economic growth on healthcare resource inequality from a longitudinal perspective [7]. This study differs from previous studies in providing evidence from the perspective of spatial externality. As shown in Table 6, the higher the non-agricultural industry rate, the better the local ULHC and LLHC. It is worth noting that the results also show that the non-agricultural industry rate had significant negative externalities on the ULHC of surrounding areas, which indicates that the aggregation effect is greater than the diffusion effect when the industrial structure is better. On the one hand, industrial structure upgrades attract investment and healthcare professionals to developed regions [53]; on the other hand, they also elevate local residents’ demands for healthcare. This reason, combined with the unordered pattern of medical treatment, means that the over-clustering of ULHC is hard to control.

The urbanisation rate has a positive impact on LLCH. A possible explanation for this might be that regions with high urbanization levels are generally economically developed and have higher amounts of LLHC. Another possible explanation for this is that living in more urbanized areas increases the risk of acquiring chronic diseases, due to the tendency of urban residents to have a worse diet and perform less physical activity than rural residents, resulting in increased LLHC demands.

### 4.4. Healthcare Expenditure

Government fiscal health expenditure, household health expenditure and social health expenditure are the three major sources of medical and healthcare funding. Current studies have provided empirical evidence that supports the presence of spillover effects from government expenditure to healthcare resources [34,54]. Political yardstick competitions and mimetic effects between governments are the major explanations of the government healthcare expenditure spillover effect [55]. The DSDM results of this study further distinguish the differences between ULHC and LLHC. They show that government healthcare investment enhances the amount of ULHC, not only in local regions but also in surrounding provinces. Nevertheless, for LLHC, government healthcare investment only improves local conditions and has no spatial spillover effects. Moreover, under constant budget constraints, the larger share of education expenditure crowds out healthcare public expenditure [13]; thus, government education expenditure has significantly negative direct and indirect impacts on both levels of healthcare in the short and long terms in this case.

Although government and social health expenditure can meet the basic medical and healthcare needs of residents, out-of-pocket (OOP) payments still account for a large proportion of total health expenditure. The results of this study show that family health expenditure had significant impacts on both levels of healthcare, which indicates that private spending may have exacerbated inequalities in resource distributions, and government investment may have failed to prevent this happening. LLHC has become increasingly dependent on government funding over the past few decades. In contrast, governmental funding support to large hospitals is limited and they often have to survive through a fees-for-services system. Medical charges are shared by social health insurance programs and OOP payments, and OOP payments still comprise a large proportion of hospital charges. Meanwhile, in this case, both rural and urban family healthcare expenditure were correlated with the amount of healthcare resources in surrounding areas. These results suggest that population mobility also makes it difficult for government investment in healthcare to achieve a balance between supply and demand [5]. Increasingly, social health insurance funds reimburse hospital expenses incurred outside of their jurisdictions due to high population mobility.

## 5. Conclusions and Policy Implications

### 5.1. Key Findings and Policy Implications

This study explored the spatio-temporal distributions of the two levels of public healthcare resources that exist in China by applying Moran’s *I* method. The influences on the two levels of healthcare and their spatial spillover effects were examined using a spatio-temporal-lagged DSDM model with FE by implementing the ML, LR, Hausman and QML estimation procedures.

There are two key findings obtained by this study. One important finding is that despite great increases in both levels of healthcare resources, significant spatial disparities remain. The distribution of ULHC and LLHC exhibited different patterns spatially, with LLHC tending to be distributed more equally. According to the spatial disparities of the two levels of healthcare resources, three stages were identified over the study period. Another interesting finding is that the DSDM analysis revealed significant direct and indirect effects at both short-term and long-term scales for both levels of healthcare resources, while the influencing factors had different impacts on the different levels of healthcare resources. In general, long-term effects were greater for ULHC and short-term effects were greater for LLHC. The spillover effects of ULHC were more significant than those of LLHC. More specifically, industrial structure, traffic accessibility, government expenditure and family healthcare expenditure were the main determinants of ULHC, while industrial structure, urbanisation, topography, traffic accessibility, government expenditure and family healthcare expenditure were the main determinants of LLHC.

The findings of this study yield the following implications for healthcare policy. First, the analysis of healthcare at its different levels is of great value to policy-makers seeking to optimize healthcare allocation in more sophisticated and systematic ways for the purposes of HDT reform. Considering that ULHCs are highly clustered and their aggregation effects are greater than their diffusion effects, policy makers should pay more attention to enhancing macro-controllability to prevent the over-scaling of large general hospitals and the over-clustering of ULHC. Mitigation measures, such as establishing cross-regional hospital consortia and counterpart support, should be implemented to promote a trickle-down effect of ULHC from developed areas to surrounding areas. Second, the findings of direct and indirect effects at the short and long terms provide evidence for policymakers seeking to mitigate spatial inequity more strategically. Spatial interdependence between regions should be fully considered for ULHC given its’ much more significant spillover effects; besides, more attention should be paid to the long-term effects of ULHC and the short-term effects of LLHC. Third, in countries like China where a large population lives in mountainous areas, the impact of topography on the spatial equity of healthcare resource should be considered. LLHC to the northeast of the Hu Huangyong Line, particularly in mountainous areas, needs to be strengthened. Complimentary assistance from developed regions and targeted healthcare professional policies should be implemented in northeastern mountainous areas to narrow the gap in LLHC between mountainous and plains areas. Fourth, the spatial spillover effects of healthcare suggest that the inter-regional connectivity of public medical insurance should be improved, considering the large interprovincial mobility of the Chinese population.

### 5.2. Research Strengths and Limitations

This study has several limitations. First, because it lacks multilevel healthcare data at the city and county scales, a micro-level analysis of the whole country could not be conducted. Thus, differences in spatial distributions and spillover effects at different spatial scales could not be compared. Second, this study focused on spatial effects resulting from the interaction of healthcare resources between provincial governments. Vertical coordination, which include the interaction of healthcare resources between different levels of government, have not been discussed.

Despite the limitations of this study, the findings have certain strengths. Firstly, recent research has applied dynamic spatial econometric models to test the spillover effects of population growth, cigarette consumption, waste disposal taxes and pensions [33]. Using the same methodology, this paper enriches the results of these empirical studies by focusing on healthcare resource distributions. By using these methods, three-dimensional analyses of public healthcare resource distributions were conducted to identify upper-level and lower-level effects, direct and indirect effects, and long-term and short-term effects. Secondly, the findings that government and individual expenditures affected the two levels of healthcare take this research field a step forward. The findings for ULHC are in line with those of Jeleskovic and Schwanebeck and Baltagi et al. – that increases in government healthcare investment in one region encourage policy makers to increase the budgets of neighbouring regions; however, we also found that these phenomena were not significant for LLHC. Furthermore, these results reinforce the finding of Zheng et al. that crowding-out effects between different kinds of public expenditure influence the outcomes of healthcare investment in local and neighbouring regions. In addition, this study verified the impacts of OOP healthcare expenditure on healthcare resources in local and surrounding regions. Thirdly, unlike Zheng et al. [13] and Yang and Zhang [19], who used wastewater and air pollution as natural explanatory variables, we used the proportion of mountainous area as a proxy variable to explain how topography influences healthcare resources. The findings verify our hypothesis that steep topography has a negative impact on local LLHC.

### 5.3. Future Research

Future research efforts could focus on more specific spatial analyses of different kinds of healthcare resources; for example, private hospitals and rehabilitation institutions, and analyse the spatial competition or complementary effects between public and private providers of healthcare services. Spatial characteristics of the multilevel healthcare system, which include community-, county-, municipal-, provincial- and state-level healthcare resources, could be further estimated. Further study with a greater focus on coherence and coordination between different levels of healthcare is, therefore, suggested. More advanced spatial analysis methodologies, such as posterior model probabilities, geographical simulation and optimization systems, could be applied to space-based research on healthcare.

## Figures and Tables

**Figure 1 ijerph-16-00582-f001:**
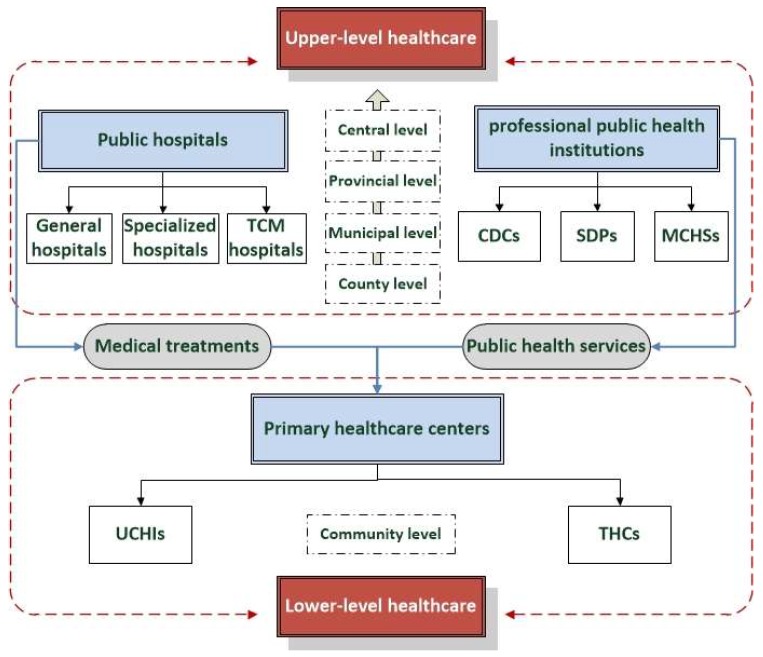
The two-level healthcare system of China. (TCM: traditional Chinese medicine; CDCs: control disease centres; SDPs: specialised disease prevention centres; MCHSs: maternal and child health stations; UCHIs: urban community health institutions; THCs: town health centres).

**Figure 2 ijerph-16-00582-f002:**
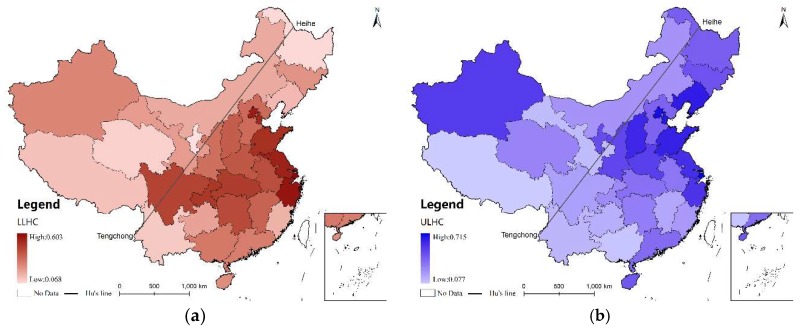
Spatial distribution of the mean values of LLHC (lower-level healthcare) and ULHC (upper-level healthcare) (2003–2015), (**a**) LLHC, (**b**) ULHC.

**Figure 3 ijerph-16-00582-f003:**
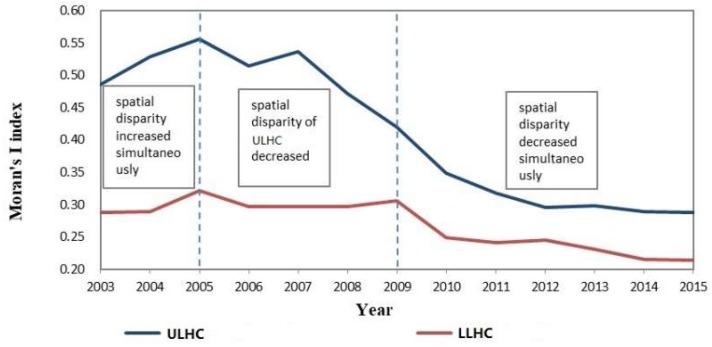
Variation in the spatial disparity of ULHC and LLHC in China.

**Figure 4 ijerph-16-00582-f004:**
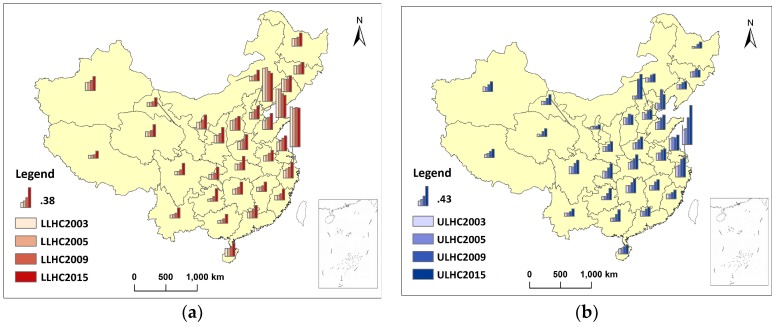
Spatio-temporal variation in characteristics of LLHC and ULHC in China, (**a**) LLHC, (**b**) ULHC.

**Table 1 ijerph-16-00582-t001:** Evaluation index of two levels of public healthcare resources.

Goal Layer (Y)	Criterion Layer (B)	Entropy Weight (w)	Index Layer (A)	Goal Layer (Y)	Criterion Layer (B)	Entropy Weight (w)	Index Layer (A)
Upper-level healthcare index (ULHC)	Upper-level medical institutions	0.453	No. of public hospitals	Lower-level healthcare index (LLHC)	Primary healthcare centres	0.510	No. of UCHIs
			No. of CDCs				
			No. of MCHSs				No. of THCs
			No. of SDPs				
	Upper-level hospital beds	0.309	No. of beds in public hospitals		Primary healthcare centre beds	0.267	No. of beds in UCHIs
			No. of beds in MCHSs				No. of beds in THCs
			No. of beds in SDPs				
	Upper-level healthcare professionals	0.237	No. of health professionals in public hospitals		Primary healthcare professionals	0.223	No. of health professionals in UCHIs
			No. of health professionals in CDCs				
			No. of health professionals in MCHSs				No. of health professionals in THCs
			No. of health professionals in SDPs				

CDCs: control disease centres; SDPs: specialised disease prevention centres; MCHSs: maternal and child health stations; UCHIs: urban community health institutions; THCs: town health centres.

**Table 2 ijerph-16-00582-t002:** Factors influencing healthcare resource distribution.

Factor Type	Proxy Variable	Factor Description	Computing Method
Socioeconomic factors	*X_NAR_*	Non-agricultural industry rate	ratio of secondary and tertiary industry to GDP (%)
	*X_UR_*	Urbanization rate	ratio of urban population to total population (%)
Geographical factors	*X_PMA_*	Proportion of mountainous areas	obtained from the Digital Mountain Map of China (%)
	*X_TA_*	Traffic accessibility	traffic network density (assign weights to railways, highways, first-class roads, second-class roads and other roads, and divide the weighted summary by the number of administrative areas)
Government investment	*X_HI_*	Healthcare investment	ratio of healthcare, social security and welfare investment to total fixed-asset investment (%)
	X*_EI_*	Education investment	ratio of education investment to total fixed-asset investment (%)
Family healthcare expenditure	*X_UFHE_*	Urban family healthcare expenditure	urban household healthcare expenditure per capita (Yuan)
	*X_RFHE_*	Rural family healthcare expenditure	rural household healthcare expenditure per capita (Yuan)

**Table 3 ijerph-16-00582-t003:** Descriptive statistics and sources of the data.

Variable	Source	Obs	Mean	Std.Dev.	Min	Max
ULHC	(CHSY) (2004–2016)	248	0.205	0.144	0.020	0.780
LLHC	(CHSY) (2004–2016)	248	0.200	0.126	0.030	0.860
X_NAR_	(CSY) (2004–2016)	248	88.162	6.044	65.780	99.560
X_UR_	(CSY) (2004–2016)	248	49.681	15.157	19.700	89.610
X_PMA_	DMMC	248	63.303	27.986	0.800	98.100
X_TA_	(CSY) (2004–2016)	248	0.264	0.143	0.030	0.620
X_HI_	(CSY) (2004–2016)	248	0.804	0.262	0.270	2.220
X_EI_	(CSY) (2004–2016)	248	1.959	0.908	0.660	6.290
X_UFHE_	(CSY) (2004–2016)	248	847.108	370.643	221.700	2464.500
X_RFHE_	(CSY) (2004–2016)	248	382.179	283.715	21.300	1395.200

Obs: observations; Std.Dev.: standard deviation.

**Table 4 ijerph-16-00582-t004:** The constraint statistics of the spatial econometric models.

ULHC			LLHC		
Test	Statistic	*p*-Value	Test	Statistic	*p*-Value
Moran’s *I*	3.219	0.0010	Moran’s *I*	2.852	0.0340
LM-error	174.562	0.0000	LM-error	37.877	0.0050
Robust LM-error	84.394	0.0000	Robust LM-error	11.582	0.0010
LM-lag	156.862	0.0000	LM-lag	21.410	0.0350
Robust LM-lag	66.694	0.0000	Robust LM-lag	5.115	0.0240
LR-error	51.295	0.0000	LR-error	65.366	0.0000
LR-lag	40.773	0.0000	LR-lag	43.012	0.0000

LM: Lagrange multiplier; LR: likelihood ratio.

**Table 5 ijerph-16-00582-t005:** Estimates of the DSDM (dynamic spatial Durbin panel models) model.

Variables	ULHC		LLHC	
	Main	Wx	Main	Wx
*L.WlnY*	0.081 (1.12)		−0.086 (−0.68)	
*lnX_NAR_*	1.504 *** (3.34)	1.545 * (1.76)	2.134 *** (2.65)	1.305 (0.78)
*lnX_UR_*	−0.085 (0.93)	0.131 (0.62)	1.177 *** (2.95)	0.056 (0.32)
*lnX_PMA_*	0.000 (0.000)	0.000 (0.000)	−0.009 (1.07)	−0.002 (0.69)
*lnX_TA_*	1.252 *** (9.35)	1.213 ** (2.42)	0.219 * (1.89)	0.228 (0.86)
*lnX_HI_*	0.042 ** (2.50)	0.108 ** (2.86)	0.161 * (1.74)	0.029 (0.76)
*lnX_EI_*	−0.043 * (−1.92)	−0.183 *** (−3.89)	−0.176 *** (−4.26)	−0.199 ** (−2.18)
*lnX_UFHE_*	−0.400 *** (−6.07)	0.393 * (2.45)	0.228 * (1.86)	−0.464 * (−1.68)
*lnX_RFHE_*	−0.011 (−0.45)	−0.102 ** (−2.47)	0.044 (0.99)	−0.123 (−1.63)
*ρ*	0.527 *** (9.67)		0.033 ** (0.38)	
*σ* ^2^	0.083 *** (14.40)		0.028 *** (14.77)	
*Adj. R* ^2^	TL: 0.177 ST: 0.636 BT: 0.332		TL: 0.817 ST: 0.841 BT: 0.393	
*LogL*	TL: −2.687 ST:169.689 BT: 117.584		TL: 178.673 ST: 188.295 BT: 106.589	

*, **, *** mean correlation is significant at the 0.10, 0.05, and 0.01 level, respectively, *t*-values in parenthesis. TL, ST, BT mean time-lagged, spatio-temporal lagged and both time and spatio-temporal lagged DSDM. Line 3–13 report the estimated parameters of the spatio-temporal lagged models.

**Table 6 ijerph-16-00582-t006:** Results of DSDM.

Variables	ULHC						LLHC					
	Short Term			Long Term			Short Term			Long Term		
	Direct	Indirect	Total	Direct	Indirect	Total	Direct	Indirect	Total	Direct	Indirect	Total
*lnX_NAR_*	1.876 ***	−4.451 **	−2.575 **	2.006 ***	−5.665 **	−3.66 **	2.143 ***	1.506	3.649	2.117 ***	1.229	3.345
	(3.39)	(−2.06)	(−2.44)	(3.35)	(−2.10)	(−2.40)	(2.69)	(0.83)	(1.64)	(2.69)	(0.74)	(1.64)
*lnX_UR_*	−0.071	0.154	0.082	−0.069	0.169	0.100	1.196 ***	0.064	1.260 ***	1.114 ***	0.041	1.155 ***
	(−0.71)	(0.36)	(0.17)	(−0.64)	(0.33)	(0.17)	(2.85))	(0.38)	(2.78)	(2.85)	(0.24)	(2.79)
*lnX_PMA_*	0.000	0.000	0.000	0.000	0.000	0.000	−0.012 *	−0.003	−0.015	−0.001	−0.002	−0.003
	(0.000)	(0.000)	(0.000)	(0.000)	(0.000)	(0.08)	(−1.73)	(−1.09)	(−0.89)	(−0.64)	(−0.69)	(−0.69)
*lnX_TA_*	1.315 ***	0.833	2.148 ***	1.356 ***	1.251 *	2.607 ***	1.264 **	0.242	1.506 ***	1.163 **	0.218	1.380 ***
	(9.34)	(1.56)	(3.59)	(9.11)	(1.86)	(3.45)	(2.44)	(1.00)	(2.82)	(2.38)	(0.89)	(2.83)
*lnX_HI_*	0.063 **	0.259 **	0.323 ***	0.070 ***	0.321 *	0.392 ***	0.170 *	0.029	0.200 *	0.157 *	0.026	0.183 *
	(2.72)	(2.49)	(2.74)	(2.80)	(2.47)	(2.67)	(1.72)	(0.77)	(1.86)	(1.70)	(0.69)	(1.87)
*lnX_EI_*	−0.075 **	−0.405 ***	−0.480 ***	−0.085 ***	−0.497 ***	−0.583 ***	−0.174 ***	−0.214 **	−0.387 ***	−0.170 ***	−0.185 **	−0.355 ***
	(−3.09)	(−4.06)	(−4.23)	(-3.29)	−(3.87)	(−4.02)	(−4.37)	(−2.12)	(-3.53)	(−4.27)	(−1.99)	(−3.56)
*lnX_UFHE_*	−0.371 ***	0.337	−0.034	−0.370 ***	0.330	−0.041	0.226 *	−0.496 *	−0.270	0.216 *	−0.483 *	−0.247
	(−5.13)	(1.16)	(−0.11)	(−4.90)	(0.94)	(−0.10)	(1.75)	(−1.76)	(−1.72)	(1.80)	(−1.81)	(−1.72)
*lnX_RFHE_*	0.004	0.188 **	0.193 **	0.008	0.225 **	0.234 **	0.043	−0.125	−0.081	0.045	−0.120	−0.074
	(0.19)	(2.44)	(2.29)	(0.35)	(2.39)	(2.26)	(0.96)	(−1.57)	(−1.11)	(0.99)	(−1.59)	(−1.11)

*, **, *** mean correlation is significant at the 0.10, 0.05, and 0.01 level, respectively, *t*-values in parenthesis.

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
