# Peer review of "Spatio-Temporal Distribution, Spillover Effects and Influences of China’s Two Levels of Public Healthcare Resources"

_ijerph, 2019, doi:10.3390/ijerph16040582_

Round 1
Reviewer 1 Report
This study examined the “Spatio-temporal Distribution and Spillover Effects” of health resources across provinces in China. I found the study interesting, but challenging to fully understand. My major concerns include: (1) the division of upper level and lower level of health institutions; and (2) the descriptions of the methods.
· Why were CDCs (Centers for Disease Control) included in ULHC? CDCs are established based on local public health needs. It is true that China is struggling in developing a HDT system for medical care. But such a HDT system (if I am allowed to use the term) has already existed. For example, consumers have to go with their registered providers for planned immunization and maternal and child health care if they want to enjoy the governmental subsidies. Spillover effects across provinces, if ever exist, would be minimal.
· The mixture of county, municipal, provincial and central levels of hospitals in ULHC may not be a good idea either. Their capacities and functions vary considerably. In addition, there exist considerable inequalities within a province.
· It is difficult to understand the spillover effects of LLHC across provinces. The definition of LLHC includes THCs for rural and UCHC for urban residents. These centers deliver both medical and public health services. Again, public health services are funded in line with their local populations and are unlikely to have any spillover effects. As noted by the authors, medical services in LLHC are usually under utilized. Patients often bypass LLHC and seek HLHC directly. Under such circumstances, it is difficult to imagine how across-province spillover effects can occur in LLHC.
· I don’t understand how the spillover effects between UCHC and LLHC were determined in this study.
· As argued by the authors, many different methodological approaches have been used in exploring spatio-temporal distributions. Please explain why Moran index was used in this study. What is the advantage of using Moran index?
· It appears that a composed health resource index incorporating number of institutions, beds and health workers was employed in this study using an entrophy-weighted method (Table 1). Please explain why and how the composed index was calculated. I noticed that hospital beds in UCHC were not considered. Many UCHCs (although not all of them) have beds. Why were these ignored?
· The spatio-temporal distributions of health resources (the composed resource index) were examined using the Moran index. I am not in a position to comment on the formula as I am not an expert in this area. However, the authors need to describe how to explain the results of the Moran index. For example, how was the conclusion of “high spatial autocorrelations” reached?
· SDM model was employed to examine the spillover effects. The authors argued that “direct impact is the average impact of an explanatory variable within the same province. An indirect effect is a measure of a spillover effect, which is the average change in a specific province's ULHC and LLHC due to explanatory variable(s) in neighbouring provinces”. Clearly, there is no examination of spillover effects between ULHC and LLHC.
· Please give a full spell of DEM when it first appears.
· The authors should provide more details of the Chinese health system in interpreting the results. For example, LLHC are increasingly dependent on government funding over the past few decades. In contrast, governmental funding support to large hospitals is limited. They have to survive through fee for services. Medical charges are shared by social health insurance programs and out of pocket (OOP) payments. OOP payments still comprise a large proportion in hospital charges. Increasingly, social health insurance funds reimburse hospital expenses occurred outside of their jurisdictions due to high population mobility.
· It is incorrect to say “that government investment do not have significant impacts on healthcare resource distribution, while household health expenditure has significant impacts on both levels healthcare”. Private spending may have exacerbated inequalities in resource distributions, and government investment may have failed to prevent this happening. But this does not mean government investment does not have a role in health resource distribution.
Author Response
Thank you very much for giving us the opportunity to revise our manuscript. We appreciated your positive and constructive comments and useful suggestions on our manuscript. They helped us to revise and improve our paper and confirm the relevance of our research. After studying the comments carefully, we made the necessary corrections, which we hope meet with your approval. The revised portions in this paper are marked in red in the paper. The main corrections in the paper and our responses to your comments are in the attached WORD file.

Reviewer 2 Report
This paper discusses relevant topics for the scientific fields. It relates to “Chinese Upper- and lower-level healthcare institutions and their spatially lagged/spillover effects”. Authors’ words suggest they can conclude “Healthcare reform, population density, elevation, industrial structure, urbanisation, education and family healthcare expenditure were the main dominant factors. Further, there were significant spatial spillover of both levels of healthcare resources. These findings have important implications for policymakers in optimising the availability of the two levels of healthcare resources.”
I realize the paper has been the result of a substantial effort from Authors. However, given the strategic role for this journal’s reviewers, I cannot provide an Acceptance letter for this version.
I suggest Authors consider the following lines as promising avenues for a significant revision of their work toward a publishable work.
Review of Literature:
It is hard to follow the (inexistent section of Review of) Literature. Additionally, Authors introduce a Table 1 which is not well explained regarding its role in the remaining discussion. I suggest Authors to organize it through a chronological line but also toward their discussion about the methodological strategy.
A Review of Literature must also address the History of the Scientific Thought of this thematic and the derived implications in terms of practical procedures and policies. The literature can be improved, namely with recent researches like those from Mourao and Vilela (‘No country for old men’? The multiplier effects of pensions in Portuguese municipalities; December 2018Journal of Pension Economics and Finance ; DOI: 10.1017/S1474747218000318) or Chen et al ( September 2017Chinese Sociological Review 49(4):293-315).
Data and Methods:
An academic paper must be a kind of repeatable path. Therefore, Authors need to significantly improve the quality of the presentation of their data. The sources must be commented as well as the sources’ limitations.
First of all, authors do not use robust methods. Therefore, what is the socio-economic relevance of discussing these concluding remarks considering a so high sample error?
Then, the empirical strategy shows there are serious problems of endogeneity in these data (denounced by the biased choice of explicative/control variables). I would suggest authors to try alternative methods in their discussion. Some remaining elements – a Posterior Matrix for selection of the Spatial Model and a discussion regarding short and long-term effects – are missing.
Conclusion and Policy Implications:
This section needs a substantial revision. Authors must clearly address the gains of this research and suggest practical innovations or policies’ changes.
The conclusions must be enriched with papers which were based on the same methodology but focused on different realities than the studied one
Overall, the English must be revised by a professional service.
Author Response

(The authors gave the same response as above.)

Reviewer 3 Report
The paper presents a summary of a research project on development of healthcare centres distribution in China. The Introduction provides the background and literature review of the problem and succeeds in arousing interest of the reader. The results are clearly presented and their description is well illustrated with figures. Section 4 provides a broad discussion of statistical results.
Section 2 Methods
The design of this chapter is confusing: a mixture of general description of the methods and measures used (2.2, 2.3.1) and direct reference to the problem in question (calculation of the problem-specific parameters in 2.1, listing factors selected as independent variables that affect the dependent variables in 2.3.2, information on the source of input in 2.4. All this information is useful, precise enough, and necessary to present the line of reasoning, but the sequence seems random. I would start with defining the variables of interest (what is going to be calculated and what for), sources of input and type of input selected for the analysis, then the methods with the steps of the analysis. Especially the steps of the analysis are missing in this chapter, though the presentation of results in Section 3 partly explains them.
In particular:
Section 2.1. I do not understand the second sentence in the first paragraph (line 100-101): Table 1 presents entropy weights related with the healthcare systems: are these values the guidelines/requirements formulated by the “healthcare systems in China”, or do they express the actual conditions for a certain date/average for the period of analysis?
What is exactly meant by ULHC and LLHC in line 101: are they indices (measures of availability of upper-and lower level health centres) or just an abbreviation for Upper Level Health Centres and Lower Level Health Centres? If they are indices, their formulas are missing (I guess they are a weighted combination of number of beds, number of doctors and other components listed in Table 1, but, as they are used as the key measure, they deserve more description, [15] and [23] do not provide it. If they are indices, are they the observed values of yit from Formula (4)? How are they related with Lit and Uit mentioned in line 129?
Section 2.2 What is Xi? Is it the observed yij of a particular year? If I is used to measure the spatio-temporal distribution of ULHC and LLHC (line 111-112), then using Xi (related later to independent variables presented much later in the text) is confusing.
I GUESS the fragment in lines 115-119 refers to the way of dividing the total period of analysis into five periods mentioned in line 186 and three periods mentioned in 198 in the Results section, but as the steps of the analysis are not explained in Section 2, it remains a guess.
Author Response

(The authors gave the same response as above.)

Round 2
Reviewer 2 Report
Now, the paper is publishable.